# On-Machine Measurement of Profile and Concentricity for Ultra-Precision Grinding of Hemispherical Shells

**DOI:** 10.3390/mi13101731

**Published:** 2022-10-13

**Authors:** Yu Wang, Chaoliang Guan, Yifan Dai, Shuai Xue

**Affiliations:** 1College of Intelligence Science, National University of Defense Technology, Changsha 410073, China; 2Laboratory of Science and Technology on Integrated Logistics Support, National University of Defense Technology, Changsha 410073, China; 3Hunan Key Laboratory of Ultra-Precision Machining Technology, Changsha 410073, China; 4State Key of High Performance Complex Manufacturing, Central South University, Changsha 410073, China

**Keywords:** on-machine measurement, hemispherical resonators, profile measurement, concentricity measurement

## Abstract

The profile and concentricity of hemispherical shells affect the frequency split and quality factor of hemispherical resonators. To compensate for machining errors caused by tool wear and tool setting, an on-machine measurement (OMM) method for the profile and concentricity of hemispherical shells in ultra-precision grinding was developed without the removal of workpieces from the machine tool. The OMM utilizes an inductive lever probe to test the inner and outer surfaces of the shell. A standard sphere is utilized to calibrate the relative position of the inductive lever probe at the two different work positions. To enhance the test accuracy of the OMM, a zero-position trigger-sampling method for the inductive lever probe was developed. It was verified to achieve a stable repeatability accuracy of 0.04 μm when using the OMM to realize a single-point sampling. Hemispherical shells were tested using the proposed OMM method. The concentricity test’s accuracy was verified to achieve accuracy better than 1 μm using a coordinate measuring machine and a standard sphere. The accuracy was 0.26 μm for testing the profiles of the hemispherical shell. The proposed OMM system was integrated with an ultra-precision machine tool. It is hoped that this method can help realize the integration function of machining-measurement-compensation.

## 1. Introduction

Hemispherical shell parts are widely used in hemispherical resonant gyroscopes [1,2] and the target capsules of inertial confinement fusion facilities [3,4]. Hemispherical shell resonators (HSRs) of Ψ-shaped fused silica are the core of the hemispherical resonant gyroscopes shown in Figure 1. The geometric defect of the HSRs causes frequency split, which leads to the failure of gyroscopes.

The key parts of HSRs are the inner and outer spheres, whose sphericity and concentricity should be controlled with high accuracy. Ultra-precision grinding can enhance the formation accuracy and reduce machine defects. Therefore, it can reduce the subsequent polishing time. When using a cup wheel for generating grinding of a spherical surface [5], two different cup wheels are required for the inner and outer spheres, leading to tool-setting errors. If a head wheel is used, dressing or replacement of the grinding wheel will also cause tool-setting errors [6]. The tool-setting error will result in concentricity error of the inner and outer spheres. In addition, no matter what grinding method is used, the wear of the grinding wheel develops with the grinding process. This affects the profile and concentricity of the inner and outer spheres [7].

To represent and compensate for the tool-setting and wear errors of grinding, it is essential to test the profile and concentricity after each grinding cycle. If the off-machine measurement is chosen, secondary clamping errors are introduced. In comparison, on-machine measurement avoids this issue and significantly improves the machining efficiency [8,9]. In addition, the excellent positioning and motion accuracy of current ultra-precision machine tools on each axis extend the measuring range and ensure invariance of the measurement datam.

OMM of the profile of HSRs is challenging because of the blocking of the surface normal caused by the anchor stem and the large curvature and sag of the HSRs. OMM is mainly categorized as contact or non-contact optical measurement, according to the type of sensors employed [10].

Optical non-contact sensors can determine the position of a point or a surface on the workpiece at high resolution and high efficiency. However, the accuracy is often sensitive to the environment and the surface roughness [11]. The Zeeko polishing machine is equipped with a Fizcam^®^ dynamic interferometer to realize OMM of mirrors under fabricated conditions [12]. However, wavefront interferometers are susceptible to vibration and air turbulence disturbances. Furthermore, the cost of commercial wavefront interferometers is commonly too high for factories. Another kind of non-contact optical sensor is the point displacement sensor. However, near-normal incidence to the object surface [13] is required. Diffuse laser triangulation sensors [14] and chromatic confocal sensors [15] can adapt to a relatively large incidence angle range. They are still difficult to arrange on an ultra-precision grinding machine for OMM of HSRs due to the blocking of the surface normal caused by the anchor stem.

Contact probes are robust to surface characteristics [16], such as roughness and shapes. Moreover, fluctuations in airflow and temperature do not influence the test results. As a representation of contact probes, 3D touch-trigger probes commonly utilized on coordinate-measuring machines (CMMs) can accommodate the complex shape and the deep sag of the HSRs. However, the commercially available 3D touch-trigger probes can only achieve trigger repeatability of around 0.25 μm [17,18]; they cannot meet the requirement of 0.1 μm repeatability. The linear variable differential transformer (LVDT) is another kind of contact sensor. It consists of three magnetic coils, as shown in Figure 2. An excitation signal is applied to the primary coil of the LVDT. The position variation of the moveable core causes a change in the voltage of the secondary coils. The stable and contactless mechanical structure enables positional measurements with excellent resolution and repeatability accuracy. Moreover, the zero-voltage position of the removable core is fixed [19]. The Moore nanotech diamond lathe is equipped with an axial-type LVDT probe. The probe shaft is supported by an air bearing that produces frictionless movement in the axial direction and generates a small measuring force [20,21]. The LVDT is promising for testing HSRs due to its excellent resolution, high repeatability accuracy, and zero position stability. However, the axial-type LVDT probe cannot be utilized for testing HSRs because of the blocking of the surface normal caused by the anchor stem.

The measurement of concentricity is performed to determine the positional relationship between the inner and outer spheres. The current methods usually simultaneously measure the inner and outer spheres in the same coordinate system. In the measurement of laser-focused capsules, Li used scanning transmission ion microscopy to measure the inner spherical contour and the concentricity of the capsule with respect to the shell, and the measurement resolution was up to the sub-nanometer level [23]. However, due to the weak penetration of the electron beam, at best, shells within micrometer thickness can be measured with this method. Ma used the differential confocal method based on ray tracing to simultaneously measure the outer surface profile and concentricity using the same instrument in the same coordinate system [4]. However, this measurement method requires the coordination of a laser interferometric measurement system and a differential confocal system. Moreover, it requires high alignment and combination between optical paths, so the system as a whole is large and is not suitable for the on-machine measurement environment. Wang used wavefront interferometry to measure the thickness errors of inner and outer surfaces using a wavelength-tuned Fizeau interferometer [24]. However, all of the above methods are offline measurements and require high reflectivity and high quality of the surface under test. OMM of the concentricity of HSRs during ultra-precision grinding with relatively low surface quality is challenging.

From the above references, the requirements of OMM for HSRs can be summarized as follows: (1) The probe is a contact-type probe equipped with a slender probe to adapt to the large sag of the inner spherical surface. (2) The probe has the ability of lateral directional measurement to adapt to the blocking of the anchor stem. (3) The accuracy for testing spherical surfaces is at the 0.1 μm level. (4) Unifying the measurement data of the inner and outer spherical surfaces of the HSRs.

To meet the above requirements, an OMM system with an XZC coordinate system configuration was built for an ultra-precision machine tool. The OMM utilized an inductive lever probe (with repeatability of 0.03 μm) to test the inner and outer surfaces of the shell. A standard sphere was utilized to calibrate the relative position of the inductive lever probe at the two different work positions. Based on the stability and repeatability of the inherent zero position of the inductive lever probe, a zero-position trigger method was proposed to overcome the linearity error of the displacement sensor and significantly improve the theoretical trigger accuracy of the contact-type OMM system. Aiming at the measurement of the concentricity of the inner and outer spherical surfaces of HSRs, a standard sphere was used to unify the coordinate system of the inner and outer spherical surfaces. The proposed OMM method can also be applied to the machining of other workpieces that need to meet the relative position accuracy of two or more complex surfaces.

The remainder of this manuscript is organized as follows: Section 2 presents the measurement system configuration and the acquisition method of the zero-position trigger that can improve the theoretical trigger accuracy of the OMM. Methods of alignment of the probe to the axis and the unifying probe coordinate systems of two probe postures are also presented; the profile and concentricity measurement procedures are detailed towards the end of this section. Section 3 describes the experimental validations and discussions. Finally, the paper is concluded in Section 4.

## 2. Methods

### 2.1. Measurement System Configuration

The proposed on-machine measurement system for the sphericity and concentricity of the HSRs is shown in Figure 3. The grinding process was carried out on a four-axis ultra-precision machine tool with an XZBC axis. The radial and axial rotation accuracy of the workpiece spindle was less than 50 nm. The straightness in each direction of the two feed axes (Z-axis and X-axis) was less than 0.005 μm/25 mm, and the positioning resolution is less than 1 nm. A lever-type inductive probe (Mahr 1318) was fixed on the B-axis of the machine tool, with repeatability of 0.03 μm. The positioning accuracy of the B-axis was ±0.1 arc seconds, and the distance between the probe and the rotation center of the B-axis was about 400 mm. It can be calculated that the positioning accuracy of the probe introduced by the rotation of the B-axis is less than 4 nm, so the influence of the angle error on the positioning accuracy can be ignored. The signal of the inductive probe was amplified by a signal amplifier (Mahr 1216) with an indication resolution of 0.01 μm and output in real time. The data acquisition card (Dewesoft) was used to synchronize the displacement signal of the X- and Z-axes of the machine tool with the probe signal and to set the trigger acquisition conditions. We prepared a standard sphere (radius of 9.49999 mm and peak-to-valley (PV) sphericity of 0.08 μm, Renishaw) that could be attached to the C-axis and easily aligned. The position of the standard sphere relative to the *Z*-axis was fixed during installation. During the measurement process of the same spherical surface, the angle between the probe and the workpiece spindle was unchanged. The spherical surface profile was measured by the touch probe by moving the X- and Z-axes of the machine tool.

### 2.2. Acquisition Method of the Zero-Position Trigger

To improve the theoretical trigger accuracy of the contact-type OMM system, we developed the zero-position trigger method of the probe, i.e., collecting the coordinates of a point when the probe indication turns to an inherent zero. In the case of only moving the *X_m_*- and *Z_m_*-axes of the machine tool, the position of the probe coordinate system relative to the machine tool coordinate system is constant, and the linearity error of the probe is not introduced. Based on the fact that the repeatable positioning accuracy of the machine tool axis (0.005 μm) is greater than the repeatability accuracy of the probe (0.03 μm), the accuracy of the value collected at a point is theoretically determined by the probe’s repeatability, i.e., 0.03 μm (ignoring the error of the probe). The detection method of the probe is to fix the feed of the *Z_m_*-axis, move the *X_m_*-axis close to the workpiece surface, and trigger to acquire the displacement signal of the *X_m_*- and *Z_m_*-axes. The trigger condition is that the voltage signal of the probe becomes zero.

An automated sampling method is proposed to improve the sampling efficiency. First, several points are manually sampled on the arc to be measured to fit the approximate position of the arc, and the number of manual sampling points is determined by the degrees of freedom of the theoretical shape of the arc. For example, for a particular meridian arc on a spherical shell, three points are manually collected to determine the position of the arc, and then the position of each nominal sampling point P_i_ is determined by interpolation. Figure 4 shows the sampling strategy of two adjacent points, where 1′–5′ is the movement path. The sampling of each point is divided into three stages: 1′–2′–3′ is the fast-approaching stage to the next sampling point after leaving the previous point, and the distance of 1′–2′ is determined by the sampling interval in the Z-direction; 3′–4′ is the actual sampling stage to detect the surface, and the range ±T of this stage is determined by the tolerance of the initial machining of this arc, i.e., the estimated range of the actual machining point of the workpiece deviating from the nominal machining point. However, the value of T must not exceed the range of the sensor, so as to prevent damage to the probe. The forward speed of the probe at this stage is generally low because the collected coordinate value between the probe reaching the zero position and the acquisition system accepting the trigger signal is impossible to fully synchronize. The faster the forward speed, the greater the delay in coordination in the trigger acquisition. Since the speed of the machine tool changes unstably during movement, this error can be considered random. Therefore, the trigger delay needs to be minimized by reducing the speed at this stage, and the specific setting of the moving speed of 3′–4′ is tested as shown in Section 3.1. The theoretical maximum trigger accuracy of the sensor is 0.03 μm, and the trigger delay should be less than this value; 4′–5′ is the probe retreat stage, 3′ coincides with 5′, and the position of 5′ indicates the end of P_i+2_ sampling and the beginning of P_i+2_ sampling.

### 2.3. Alignment of Probe

From the analysis above we can see that the repeatability of the on-machine measurement system in this paper is determined by the repeatability of the probe. Equally important is the positioning accuracy of the measurement system. We expect to obtain the accurate coordinates in the machine tool coordinate system, so the alignment height (*Y_m_*-direction) between the center of the probe sphere and the C-axis of the machine tool is the key to positioning accuracy. 

We used a standard wedge with an inclination angle of about γ = 20°and PV sphericity of less than 0.1 μm with six sides, as shown in Figure 5. First, we used the side of the inclined plane as the benchmark to ensure that the inclined plane was parallel to the *X*-axis of the machine tool and that the parallelism was within 1 μm/50 mm, and we set the rotation angle of the block to 0°. As shown in Figure 5, ∆*z* is the difference in the probe position when the *C*-axis is at 0° and 180°. The relationship between ∆*z* and the probe height error ∆*h* is as follows:(1)$$\Delta h=\frac{\Delta z}{2\sin\gamma }$$
where ∆*z* can be adjusted to within 0.1 μm so that the height error ∆*h* can be controlled within 0.2 μm. In this way, the measurement error of the probe caused by the height error is theoretically less than 0.01 μm.

### 2.4. Unifying the Probe Coordinate Systems

As shown in Figure 3, two probe postures are required to measure the inner and outer spherical surfaces to avoid probe interference. These two postures are set by the B-axis of the machine tool. In order to measure the concentricity of the inner and outer spherical surfaces, it is necessary to unify the coordinate system of the two probe postures. Before the workpiece blank is installed, we used the standard sphere to calibrate the coordinate system of the probe to the center of the C-axis, as shown in Figure 6a. We installed a standard sphere on the C-axis vacuum chuck, and we manually adjusted the position of the standard to be centered on the C-axis so the eccentricity could be controlled to less than 1 μm. Due to the eccentricity error of the standard sphere mounted on the C-axis, the position of the C-axis’ center needs to be calculated by measuring the two meridians of the standard sphere at 0° and 180°. Figure 6b shows a standard sphere of the XY cross-section on the C-axis. The probe center aligns the standard sphere’s center in the Y-direction. The two meridians in the XY cross-section are M_1_ and M_2_, respectively. Finally, the fits using least squares to determine the location of the sphere’s center are *O*_*c*1_ (*x*_*c*1_, *x*_*c*1_) and *O*_*c*2_ (*x*_*c*2_, *x*_*c*2_). The coordinates of the C-axis center are as follows:(2)$$O_{c}=\left(O_{c1}+O_{c2}\right)/2.$$

After separating the eccentricity error of the sphere relative to the C-axis, we obtained the coordinates *O**_cs_*_1_ (*x_cs_*_1_, *z_cs_*_1_*)* of the center of the standard sphere in the S_1_CS coordinate system at Posture 1. Similarly, we obtained the coordinates *O**_cs_*_2_ (*x_cs_*_2_, *z_cs_*_2_) of the center of the standard sphere in the *S_2_CS* coordinate system at Posture 2. The coordinate difference between the two centers is the coordinate difference between the origins of the two postures of the probes:(3)$$\Delta C=\left(x_{cx2}-x_{cx1},z_{cx2}-z_{cx1}\right)$$

In this way, any two probe coordinate systems can be unified into the same coordinate system. The uniformity provides the basis for measuring the concentricity of the hemispherical shell.

### 2.5. Measurement Procedure for Profile and Concentricity

The inner and outer spherical profiles are separately measured and integrated through the meridians and parallels. As shown in Figure 7a, the discrete circle points are the sampling trajectory of the meridian. Using the zero-position trigger method proposed in Section 2.2 of this paper, we needed to collect dozens of points on the meridian, providing a common datum for the subsequent measurement of the parallels. We measured about two-thirds of the spherical range of the hemisphere, considering the interference of the support rods of the hemispherical resonators. The continuous circular lines in Figure 7a indicate the sampling trajectory of the parallels. The scanning method continuously samples the parallels, and the sampling density can be increased to several kHz, offering a comprehensive shape of the circumferential profile. In theory, the zero-position trigger does not introduce linearity error of the probe, while the scanning measurement does. However, the high-precision spindle rotation of machine tools is at the level of 10 nanometers, and the form error of the machined parts in the radial direction is generally within micrometers. In the scanning measurement, the linearity error of the probe at the micron level is almost negligible. Under the premise of not reducing the theoretical measurement accuracy, the parallel scanning measurement improves the measurement efficiency and sampling density. In addition, the C-axis rotates at a constant linear velocity during latitudinal scanning to homogenize the sampling density of the entire sphere. Furthermore, in the parallel scanning measurement, what cannot be ignored is the directional inconsistency between the stylus axis and the error to be measured. There is a proportional coefficient between the probe indication value and the normal runout value of error. As shown in Figure 7b, the actual normal runout error of the workpiece surface is as follows:(4)$$h\left(\theta \right)=\frac{r_{0}\left(\theta \right)}{\cos\gamma \cdot \cos\varphi }=k_{1}\cdot r_{0}(\theta ),$$
where the roundness runout directly measured by the probe is *r*_0_ (*θ*), *θ* is the latitudinal circumference angle, *γ* is the angle between the normal direction of the workpiece contact point and the C-axis, *φ* is the angle between the stylus and the C-axis, and *k*_1_ is the correction factor for roundness measurement. The *γ* value of a point is calculated by the normal equation on the fitted curve of the point after the meridian measurement.

The concentricity of the inner and outer spherical surfaces is obtained by calculating the relative positions of the two spherical centers after measuring the profile of the inner and outer spherical surfaces. As shown in Figure 8, the coordinates of the point on the outer spherical surface of the workpiece in the *S_1_CS* coordinate system of Posture 1 are *P_i_*_,1_ (*x_i_*_,1_, *z_i_*_,1_). The coordinates of the points on the inner spherical surface in the *S*_2_*CS* coordinate system of Posture 2 are *Q_j,_*_2_ (*x_j_*_,2_, *z_j_*_,2_). According to Equation (3), we can transform the points of the inner sphere from the coordinate system of Posture 2 to Posture 1:(5)$$Q_{i,1}=Q_{j,2}-\Delta C,$$
where *i* = 1, 2, ..., m, *j* = 1, 2, ..., n. m, n is the number of sampling points, and ∆*C* is calculated by Equation (4). We fitted the measurement point sets *P_i_*_,1_ and *Q_j_*_,1_ of the inner and outer spheres, respectively, using the least squares method. Next, we obtained the best-fitting sphere center position of the inner and outer spheres and calculated the concentricity.

## 3. Results and Discussion

### 3.1. Repeatability of OMM Systems

The triggering of the probe is a process of contact and even friction with the workpiece surface. The instability of the contact force may introduce measurement uncertainty. The OMM is performed during a machining interval, so environmental changes, temperature fluctuations, coolant effect, airflow, and vibrations can all cause a shift in the zero position. We expect that after each calibration the zero position can be held for as long as possible to reduce the number of calibrations. Therefore, some testing of the repeatability of the measurement system is required.

#### 3.1.1. Repeatability Testing of Single Points

For the same measured point, we carried out multiple rounds of repeated trigger acquisition tests. Since there is a certain processing time between each OMM, another round of testing was performed every half an hour, 20 times per round, and the average and standard deviation of these 20 trigger coordinate values were calculated. As shown in Figure 9, the standard deviation of each round of testing can be considered as the short-term repeatability of the measurement system, and the standard deviation of each round of testing is within 0.04 μm. The long-term repeatability of the measurement system can be inferred from the change in the average value of each test. Within 4 h, the repeatability accuracy of the measurement system was 0.18 μm. The short-term repeatability accuracy of the measurement system was almost the same as the factory repeatability (0.03 μm) parameter of the probe, verifying the theoretical accuracy assumption of the proposed zero-position trigger method. Due to the complex environment of machine tools, the long-term repeatability of the probe can be affected by temperature fluctuations and vibration.

#### 3.1.2. The Setting of the Trigger Speed of the Moving Axis

In order to reduce the uncertainty caused by the trigger delay in the 3′–4′ stage in Section 2.2, we set a series of trigger speeds of the moving axis for trigger repeatability testing, taking the repeatability accuracy of 0.04 μm of the measurement system as the limit to find the maximum trigger speed. As shown in Figure 10, the average value and repeatability of the trigger coordinate values were tested 20 times at each speed. Zero trigger speed means that the measurement system is triggered statically, i.e., the machine tool is manually controlled to reach the zero position of the probe and keep it still. Then, the coordinate value of the moving axis of the machine tool is recorded. It can be seen that when the trigger speed is less than 15 μm/min, the trigger repeatability of the probe is not significantly affected. When the trigger speed is more than 15 μm/min, the influence of the trigger delay begins to appear, and the trigger coordinate’s value is larger than the static trigger value. After the triggering speed is more than 20 μm/min, the triggering repeatability gradually deteriorates. Therefore, in order to prevent the moving trigger value from exceeding the range of 0.04 μm of repeatability and to maximize the trigger speed, the trigger speed of 10 μm/min was selected.

Theoretically, the trigger delay should be a definite value when the condition of the measurement system remains unchanged. If this value can be calibrated out, the trigger speed can be further increased. However, the current trigger repeatability deteriorates practically after 20 μm/min, which may be caused by the instability of the trigger speed. The instability results from the acceleration–deceleration of the moving axis in each measurement cycle in Section 2.2. This is the aspect that we need to optimize further.

### 3.2. Accuracy Test of OMM

#### 3.2.1. Measurement Accuracy of the Profile

According to the calibration of the standard sphere in Section 2.4, we can determine the coordinate value of the measured point relative to the center of the C-axis under any probe posture. In order to test the measurement accuracy of the measuring system to the spherical profile, we used another standard sphere for testing based on the calibrated coordinate system of the first standard sphere. According to the measurement method described in Section 2.5, a standard sphere with a radius of 12.4999 mm and a sphericity error of 0.08 μm was used for profile measurement. We triggered and sampled 20 points for a meridian and scanned a parallel at the position of each meridian point. In order to simulate the measurement range of the hemispherical resonators, the measurement range of the meridian direction of the sphere was taken as three-quarters of the hemisphere. The measurement results are shown in Figure 11 as a three-dimensional diagram after fitting the profile measurement results, and the color represents the deviation from the ideal sphere. A total of 20 profile measurements were carried out, and all of the PV sphericity measurement results were within 0.26 μm. The PV profile was 0.26 μm. It can be approximated that the profile of the standard sphere is perfect, and that the profile measurement error of OMM is 0.26 μm. Figure 12 shows the profile measurement setup of standard sphere.

From the error distribution in Figure 11, we can see that the measurement error of the spherical profile mainly comes from the meridian direction. The process of parallel measurement only involves the C-axis rotation, so the measurement process is relatively stable. The measurement in the meridian direction requires the joint movement of the *X*-axis and the *Z*-axis and has a triggering process, representing the primary error sources of the measurement system. In addition, during the measurement of the meridian of the spherical profile, the contact point between the stylus ball and the workpiece varies, so the sphericity error of the stylus ball also introduces measurement errors, constituting a target for improving the measurement accuracy in the future. In addition, according to the factory technical data of the probe, the zero-point force of the probe is about 0.25 N. It is easy to calculate this to show that the influence of the deformation of the probe caused by contact force can be ignored.

#### 3.2.2. Measurement Accuracy of Concentricity

On the basis of the measurement of the inner and outer spherical surfaces, the concentricity of the inner and outer spherical centers is calculated based on the calibration results in Section 2.4. Figure 13 shows the profile and concentricity measurement of a HSRs. A total of 10 hemispherical resonators were processed, and the concentricity measurement results of the inner and outer spherical surfaces of the hemispherical resonators were verified using a coordinate-measurement machine (CMM) (Zeiss, Contura 7/7/6RDS) after OMM. The measurement method of the CMM was global discrete sampling, and its maximum 3D sampling accuracy was 1.5 μm. A total of 10 workpieces were machined. Each workpiece was measured five times on the machine for concentricity in the *Z*-axis direction and the radial concentricity of the XY plane. The average of five measurements for each workpiece is listed in Table 1. The deviation between the radial concentricity results and the CMM results was within 0.3 μm, and the deviation between the *Z*-axis concentricity and the CMM results was within 1 μm. The fact that the workpiece is formed by the C-axis rotation meant that the geometric center of surface of the HSRs was almost near the C-axis. Thus, the radial concentricity was always small for HSRs. Generally, the axial triggering accuracy of the CMM probe is lower than that of the radial direction, so the axial concentricity measurement accuracy of the CMM is significantly different from that of the OMM

The concentricity measurement accuracy of OMM may be higher than that of the CMM, resulting in the measurement deviation of the concentricity between the CMM and the OMM. At present, we do not have standard instruments with higher accuracy than the CMM to measure the actual concentricity of the hemispherical resonator, so we used an indirect method for verification. Consistent with the standard sphere used in Section 3.2.1, the standard sphere was used to simulate the measurement of the spherical shell without loss of generality, dividing the standard sphere into two parts, as shown in Figure 6a: the left hemisphere was measured by the outer spherical probe, and the right hemisphere was measured by the inner spherical probe. The left and right hemispheres of a standard sphere can be considered concentric. We used the OMM system to simulate the concentricity test of the left and right spherical surfaces of the standard sphere. A total of 20 simulation tests of inner and outer spherical probes were carried out. The measurement results of the concentricity of the left and right hemispheres of the standard sphere were all less than 1 μm. Therefore, we have reason to believe that the OMM system can measure concentricity with an accuracy of 1 μm.

## 4. Conclusions

In this paper, an OMM method with an inductive lever probe integrated into an ultra-precision grinding machine tool was proposed. The OMM utilized an inductive lever probe (with repeatability of 0.03 μm) to test the inner and outer surfaces of the shell. A standard sphere was utilized to calibrate the relative position of the inductive lever probe at the two different work positions. To enhance the test accuracy, a zero-position trigger method for the inductive probe was proposed to provide the OMM system with a short-term repetitive trigger acquisition accuracy of 0.04 μm and a long-term trigger acquisition accuracy of better than 0.2 μm. The proposed OMM method was verified by installing a standard sphere at the workpiece position of the machine tool and using a traditional coordinate measurement machine. The measurement results show that the established OMM system provides accuracy of better than 0.3 μm for the profile measurement of the spherical shell. By unifying the two coordinate systems of the probe at two working positions, the concentricity measurement accuracy was enhanced to be better than 1 μm for the spherical shell. This measurement system has significant advantages for small, deep, concave spherical shells—especially for hemispherical resonators.

## Figures and Tables

**Figure 1 micromachines-13-01731-f001:**
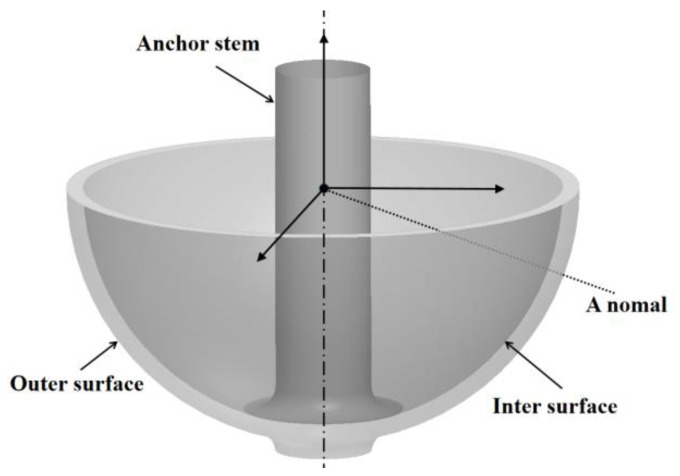
Structure of a hemispherical shell resonator.

**Figure 2 micromachines-13-01731-f002:**
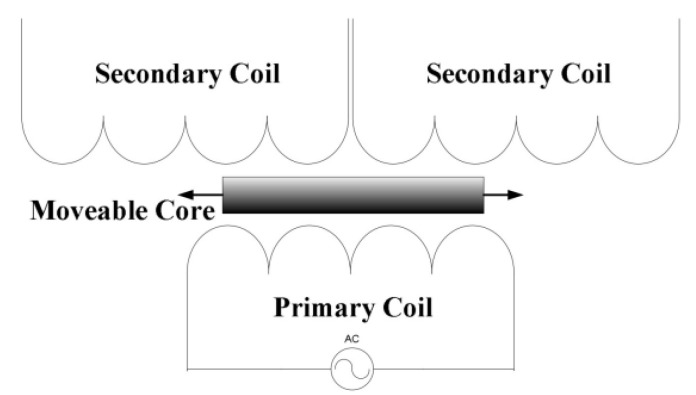
The working principle of LVDTs [22].

**Figure 3 micromachines-13-01731-f003:**
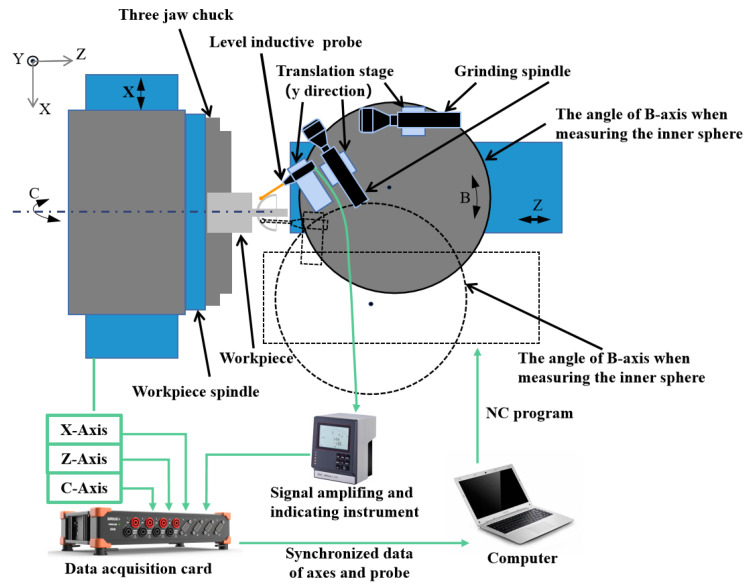
On-machine measurement system for the HSRs and signal flow.

**Figure 4 micromachines-13-01731-f004:**
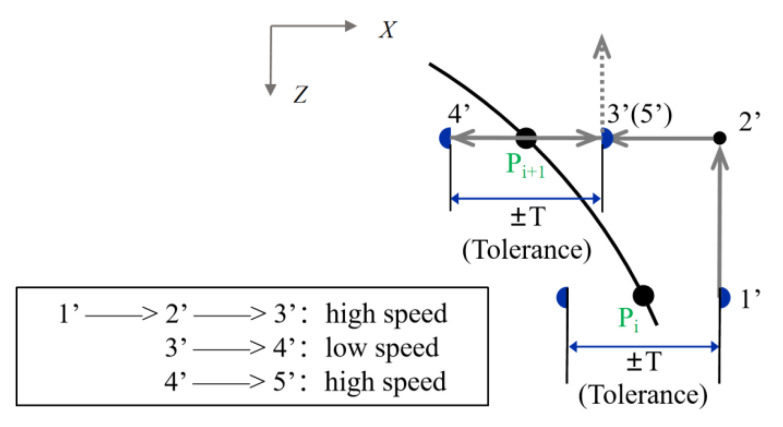
Sampling trajectory planning for two adjacent points.

**Figure 5 micromachines-13-01731-f005:**
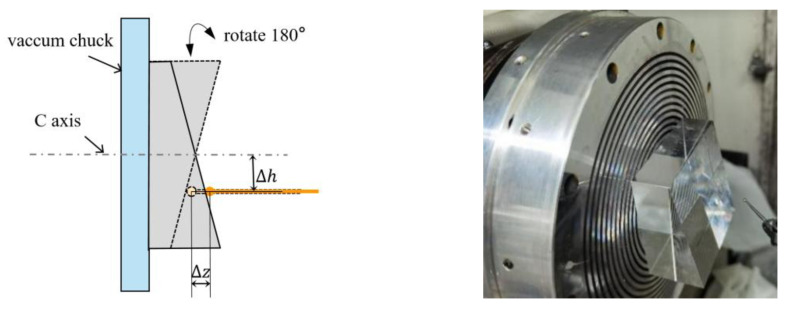
Height alignment of the probe with a standard wedge.

**Figure 6 micromachines-13-01731-f006:**
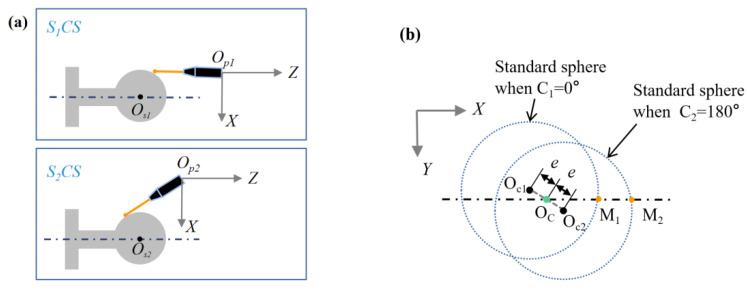
(**a**) Unifying two probe coordinate systems with the centered standard sphere. (**b**) Influence of the standard ball installation eccentricity.

**Figure 7 micromachines-13-01731-f007:**
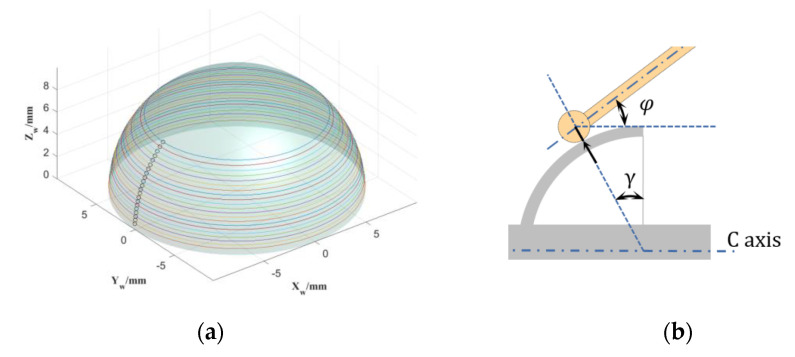
(**a**) Sampling trajectory for separating warp and weft; (**b**) correction of probe leverage.

**Figure 8 micromachines-13-01731-f008:**
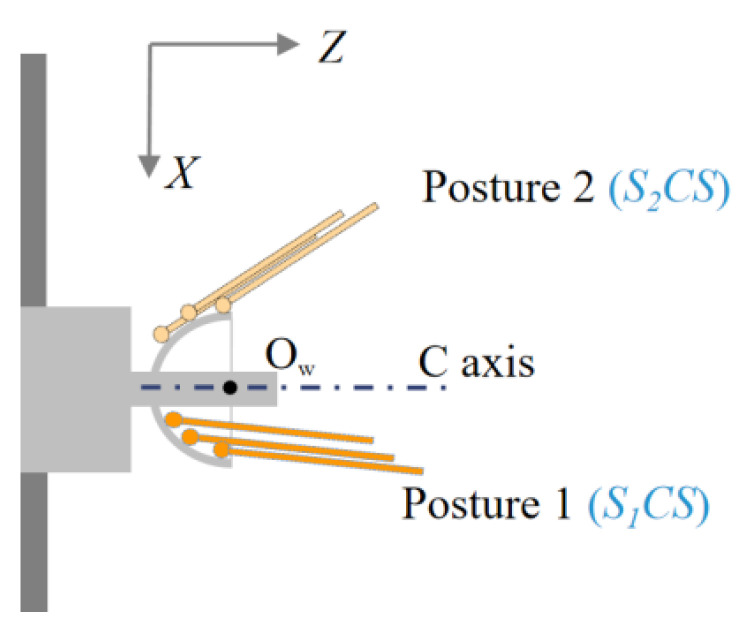
Measurement process for the concentricity of the inner and outer spherical surfaces.

**Figure 9 micromachines-13-01731-f009:**
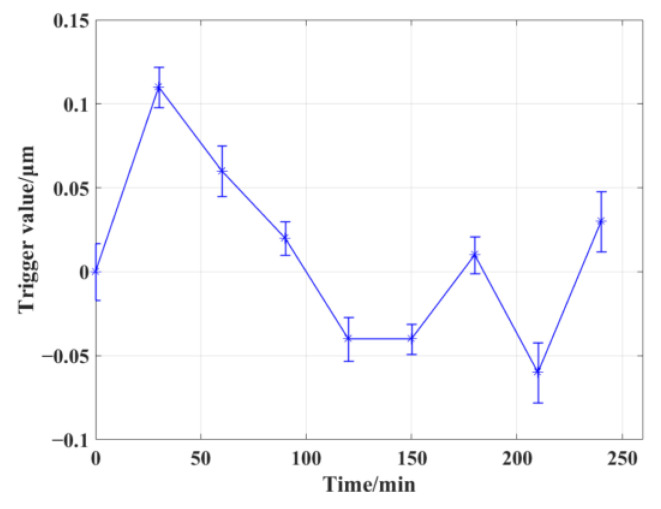
Long-term repeatability testing of the OMM.

**Figure 10 micromachines-13-01731-f010:**
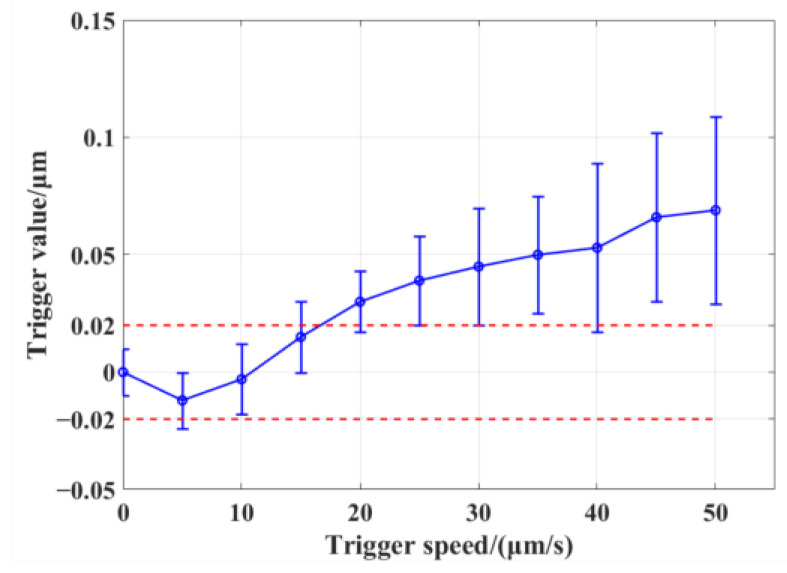
The relationship between the trigger speed of the moving axis and the coordinate value collected by the moving axis.

**Figure 11 micromachines-13-01731-f011:**
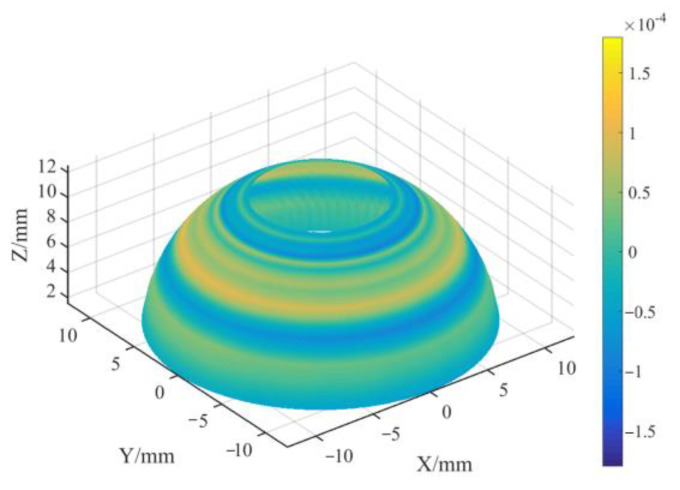
Local profile measurement results of a standard sphere.

**Figure 12 micromachines-13-01731-f012:**
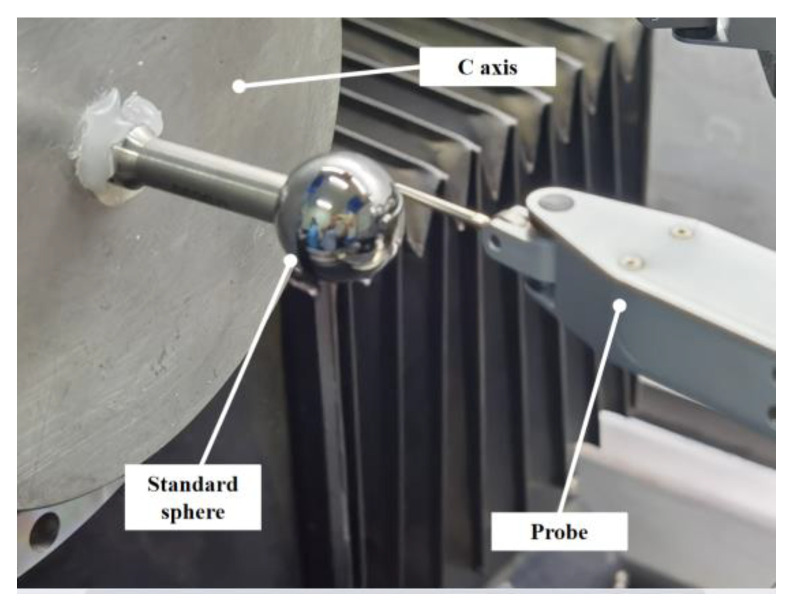
Profile measurement setup of a standard sphere.

**Figure 13 micromachines-13-01731-f013:**
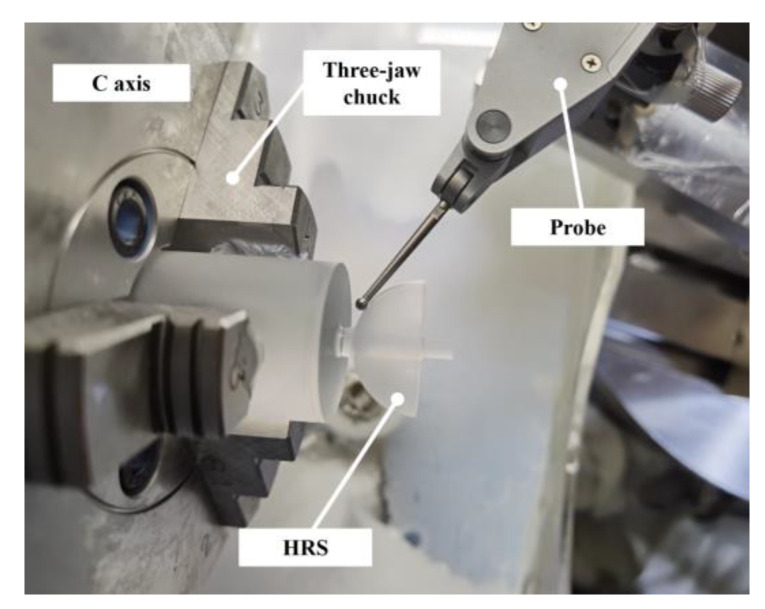
Profile and concentricity measurement of an HSR.

**Table 1 micromachines-13-01731-t001:** Concentricity test results of 10 HSRs.

	Workpiece No.	1	2	3	4	5	6	7	8	9	10
**Axial concentricity/μm**	OMM	1.31	−1.55	1.22	−2.55	3.52	1.95	6.2	2.92	−3.02	3.83
	CMM	2.47	−2.23	2.79	−0.84	3.25	2.98	5.82	2.91	−3.91	2.36
**Radial concentricity/μm**	OMM	0.23	0.11	0.33	0.45	0.22	0.25	0.15	0.52	0.35	0.42
	CMM	0.28	0.34	0.54	0.15	0.22	0.30	0.33	0.35	0.39	0.68

## Data Availability

The data presented in this study are available upon request from the corresponding author. The data are not publicly available because they are part of an ongoing study.

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
