# Peer review of "On-Machine Measurement of Profile and Concentricity for Ultra-Precision Grinding of Hemispherical Shells"

_micromachines, 2022, doi:10.3390/mi13101731_

Round 1

Reviewer 1 Report

The authors described an OMM method for measuring the profile and concentricity of hemispherical shells of ultra-precision grinding machine tools. An inductive lever probe is integrated, and a corresponding zero position trigger method is also proposed to improve the measurement accuracy. Verification experiments are designed for the measurement of the profile and concentricity of HSRs.

As my questions and comments are addressed, I would recommend that this manuscript can be accepted.

1. Do the feed axes used by the system have angular errors? How much is the impact on measurement and coordinate positioning?

2. The model of Mahr probe, signal amplifier and Renishaw standard ball used in the system should be given.

3. Line 163-166:

  The repeatable positioning accuracy of the machine tool axis is 1/2 of the repeatability accuracy of the probe, which is not much less than the repeatability accuracy of the probe. Is it appropriate to directly ignore the error of the moving axis? And the contents in brackets on line 166 are incorrect.

4. During the scanning measurement, will the contact force and friction between the probe-tip and the measured circle cause bending and deformation of the probe shaft? What is the sphericity of the probe-tip?

5. The manuscript have to be carefully checked to avoid mistakes in details. Such as,

  “Prove” in Figure 3 is misspelled, and “inner sphere” appears twice.

   Line 37: “sphere” should be plural.

   What is the full spelling of PV? It should be written clearly when used for the first time.

   Line 197: where is “2.4.1”?

   Line 266: “Figure 6(b)” should be replaced by “Figure 7(b)”.

   Line 393: where is “Fig 5(a)”?

6. Grammar mistakes exist in the manuscript. The English writing should be improved.

Reviewer 2 Report

The authors developed an on-machine measurement method for profile and concentricity of hemispherical shell in 16 ultra-precision grinding. The research is timely and novelty. The data is abundant and well presented. The manuscript is well written. I recommend to publish this manuscript after minor revision.

1. Cited references could be improved.

2. The data such as, “repeatability”, “accuracy”, should be listed carefully. How many sets of individual experiments were used in the statistical calculation?

Round 2

Reviewer 1 Report

The authors have fully considered and carefully revised my comments, and I suggest that this  manuscript can be accepted.